# The Novel *h*DHODH Inhibitor MEDS433 Prevents Influenza Virus Replication by Blocking Pyrimidine Biosynthesis

**DOI:** 10.3390/v14102281

**Published:** 2022-10-17

**Authors:** Giulia Sibille, Anna Luganini, Stefano Sainas, Donatella Boschi, Marco Lucio Lolli, Giorgio Gribaudo

**Affiliations:** 1Department of Life Sciences and Systems Biology, University of Torino, 10123 Torino, Italy; 2Department of Sciences and Drug Technology, University of Torino, 10125 Torino, Italy

**Keywords:** influenza virus, host-targeting antivirals, de novo pyrimidine biosynthesis, dihydroorotate dehydrogenase, MEDS433, dipyridamole, combination treatment

## Abstract

The pharmacological management of influenza virus (IV) infections still poses a series of challenges due to the limited anti-IV drug arsenal. Therefore, the development of new anti-influenza agents effective against antigenically different IVs is therefore an urgent priority. To meet this need, host-targeting antivirals (HTAs) can be evaluated as an alternative or complementary approach to current direct-acting agents (DAAs) for the therapy of IV infections. As a contribution to this antiviral strategy, in this study, we characterized the anti-IV activity of MEDS433, a novel small molecule inhibitor of the human dihydroorotate dehydrogenase (*h*DHODH), a key cellular enzyme of the de novo pyrimidine biosynthesis pathway. MEDS433 exhibited a potent antiviral activity against IAV and IBV replication, which was reversed by the addition of exogenous uridine and cytidine or the *h*DHODH product orotate, thus indicating that MEDS433 targets notably *h*DHODH activity in IV-infected cells. When MEDS433 was used in combination either with dipyridamole (DPY), an inhibitor of the pyrimidine salvage pathway, or with an anti-IV DAA, such as N^4^-hydroxycytidine (NHC), synergistic anti-IV activities were observed. As a whole, these results indicate MEDS433 as a potential HTA candidate to develop novel anti-IV intervention approaches, either as a single agent or in combination regimens with DAAs.

## 1. Introduction

Influenza remains a major public health challenge. Every year around the world, influenza viruses (IVs) in fact cause approximately one billion infections, with 3–5 million of severe related respiratory complications, which result in 290,000–650,000 deaths among high-risk groups, with an even greater impact in developing countries [1,2,3,4].

Seasonal vaccines represent the most effective measure for the prevention and control of IV infections [5]. Nevertheless, vaccines do not allow sufficient protection to alleviate the annual impact of IVs [6], and thus, the current intervention strategies rely also on antiviral agents to reduce the burden of complications and case fatality rates. In this regard, three classes of direct-acting antiviral (DAA) drugs have been approved: amantadanes, neuraminidase inhibitors (NAIs), and RNA-dependent RNA polymerase (RdRp) complex inhibitors [2,3,4,7]. Amantadanes act by blocking the IAV M2 ion channel; however, due to resistance by essentially all circulating IVs, they are no longer recommended [8]. Thus, NAIs, such as peramivir, zanamivir, and above all, oseltamivir, represent the standard-of-care for therapeutic management of IV infections [9]. NAIs have been employed successfully for two decades; however, during 2007–2009, resistance to oseltamivir rose drastically among seasonal H1N1 IAV due to the appearance of the NA H275Y amino acid substitution [10]. The subsequent global spread of the 2009 H1N1 IAV pandemic strain, devoid of this mutation when it emerged, reduced the frequency of NAI resistance in seasonal IV to low levels (<2%), and since then, it has remained low [11]. However, the rapid emergence of IVs with reduced inhibition by oseltamivir between 2007 and 2009 indicated that resistance to NAIs can emerge and spread among circulating IVs [12].

More recently, RdRp inhibitors, such as baloxavir marboxil (baloxavir), which targets the PA subunit of RdRp, and favipiravir, an inhibitor of the PB1 subunit, have been licensed for the treatment of uncomplicated IV infections and those resistant to other antivirals, respectively [13,14]. Globally, IV variants showing reduced susceptibility to baloxavir are detected with low frequency [11].

The available data, however, suggest that the genetic barrier against viral resistance to some approved DAA may be low and pose challenges in the control of IV infections by a curative approach, thus making obvious the need to develop alternative anti-IV agents characterized by new mechanisms of action, with a low propensity to drive the selection of resistant strains, concurrently effective against antigenically different IVs, and thus, promptly deployable against new zoonotic highly pathogenic IVs that may emerge in the future in human populations [12,15].

Taking into consideration these requirements, small molecules able to interfere with those cellular factor and biochemical pathways essential for IV replication may be considered as compelling alternatives to the de novo development of DAAs, inasmuch as such host-targeting antivirals (HTAs) may offer both a broad-spectrum of activity against different IVs and a high genetic barrier against the development of IV resistance [16].

Pyrimidine nucleotides’ availability in infected cells is crucial for efficient virus replication, and thus, compounds targeting the cellular pathways responsible for providing adequate supply of pyrimidines have the potential to be developed as effective HTA agents [17]. Especially, in metabolically quiescent uninfected airway epithelial cells, in which productive IV replication occurs, the pyrimidine demands are fulfilled through the salvage pathway from intracellular nucleic acid degradation and from the import of extracellular nucleotides in the bloodstream [17]. In contrast, in virus-infected cells, including those infected with IV, to keep up with the high pyrimidine demands required for viral gene expression and replication, the de novo pyrimidine biosynthesis pathway is up-regulated [17,18,19,20,21]. In this biochemical pathway, the human dihydroorotate dehydrogenase (*h*DHODH) catalyzes the oxidation of dihydroorotic acid (DHO) to orotic acid (ORO), a rate-limiting step of in the biosynthesis of uridine and cytidine required to fulfil the cell’s pyrimidine nucleotide demand [22,23]. Thus, given its critical role in virus-infected cells, while being dispensable in uninfected cells, *h*DHODH can be considered a druggable target of choice for the development of HTAs [17,22].

To contribute to this antiviral strategy, in the last few years, we synthesized a new class of small molecules, *h*DHODH inhibitors [24,25,26], which were designed on the scaffold of brequinar, one of the most-potent *h*DHODH inhibitors developed so far [27] and for which an anti-IV activity was recently observed [28]. Among our new *h*DHODH inhibitors, MEDS433 proved be the most effective at inhibiting the in vitro enzymatic activity (IC_50_ 1.2 nM) and at binding in the ubiquinone binding site of *h*DHODH in co-crystallization experiments [24]. Therefore, it was chosen to investigate its suitability as a new HTA [29]. As a confirmation of this hypothesis, recently, we reported the ability of MEDS433 to inhibit the in vitro replication of herpes simplex virus type 1 (HSV-1) and type 2 (HSV-2) [30], as well as of human coronaviruses (CoVs), such as the prototypic α-hCoV-229E and the β-CoVs hCoV-OC43 and SARS-CoV-2 [31].

Based on these premises, the aim of this study was to expand the potential of MEDS433 as a broad-spectrum antiviral by investigating its anti-IV activity. We report the characterization of the ability of MEDS433 to inhibit the in vitro replication of both IAV and IBV as a consequence of a selective block of *h*DHODH enzymatic activity, as well as its suitability for combination treatments with other anti-pyrimidines compounds and with DAAs. These results suggested MEDS433 as a promising HTA candidate, as either a single agent or in combination regimens, to design new therapeutic strategies for the treatment of IV infections.

## 2. Materials and Methods

### 2.1. Compounds

MEDS433 was synthesized as described previously [24]. Brequinar, uridine, cytidine, orotic acid (ORO), dihydroorotic acid (DHO), dipyridamole (DPY), and N^4^-hydroxycytidine (NHC or EIDD1931) were obtained from Sigma-Aldrich (St. Louis, MO, USA). All compounds were resuspended in DMSO.

### 2.2. Cells and Viruses

The Madin–Darby canine kidney (MDCK) (ATCC CCL-34), the human adenocarcinoma alveolar basal epithelial A549 (ATCC CCL-185), and the human lung adenocarcinoma Calu-3 (ATCC HTB-55) cell lines were purchased from the American Type Culture Collection (ATCC) and cultured in Dulbecco’s modified Eagle medium (DMEM; Euroclone (Milan, Italy)) supplemented with 10% fetal bovine serum (FBS, Euroclone), 2 mM glutamine, 1 mM sodium pyruvate, and 100 U/mL penicillin and 100 μg/mL streptomycin sulfate (P/S, both from Euroclone). All IV infections were performed in the presence of 2 μg/mL of trypsin TPCK treated by bovine pancreas (Sigma-Aldrich) and 0.14% of bovine serum albumin (Sigma-Aldrich).

The Influenza A virus strain A/Puerto Rico/8/34 (PR8) H1N1 (IAV) and influenza B virus strain B/Lee/40 (IBV) were a generous gift from Arianna Loregian (University of Padua, Padua, Italy). IAV and IBV were propagated and titrated by the plaque assay on MDCK cells, as previously described [32,33].

### 2.3. Cytotoxicity Assay

Cultures of MDCK, A549, or Calu-3 cells, seeded 24 h before in 96-well plates (10,000 cells/well), were exposed to increasing concentrations of the vehicle (DMSO) or of different compounds. After 48 h of incubation, the 3-(4,5-dimethylthiazol-2-yl)-2,5-diphenyltetrazolium bromide (MTT) method [34] was employed to determine the number of viable cells.

### 2.4. Antiviral Assays

The antiviral activity of MEDS433 or brequinar was determined by the plaque reduction assay (PRA) in MDCK cells or by the virus yield reduction assay (VRA) in A549 cells. For the PRA, MDCK cells were seeded in 24-well plates and, after 24 h, exposed 1 h prior to infection to increasing concentrations of MEDS433 or brequinar and then infected with IAV or IBV (50 PFU/well). Following virus adsorption (1 h at 37 °C), cultures were maintained in medium containing the corresponding compounds, 2 μg/mL of trypsin TPCK, 0.14% of bovine serum albumin, and 0.7% Avicel (FMC BioPolymer (Philadelphia, PA, USA). At 48 h post-infection (h p.i.), the cell monolayers were fixed with 4% formaldehyde-phosphate-buffered saline for 1 h at room temperature (RT) and stained with a solution of crystal violet and 20% ethanol. The viral plaques were microscopically counted, and the mean plaque counts for each drug concentration are expressed as a percentage of the mean plaque counts of control virus (DMSO). The GraphPad Prism software version 8.0 was used to determine the concentration of compounds that produced 50 and 90% reductions in plaque formation (EC_50_ and EC_90_). For the VRA, A549 or Calu-3 cells seeded in 24-well plates were treated with increasing concentrations of MEDS433 or brequinar and then infected with IAV or IBV at an MOI of 0.001 PFU/cell. After virus adsorption, cells were incubated in medium containing the corresponding compounds, 2 μg/mL of trypsin TPCK and 0.14% of bovine serum albumin. At 48 h p.i., the cell supernatants were harvested and IV yield titrated on MDCK cells.

For time-of-addition experiments, MDCK cells were seeded in 6-well plates and after 24 h exposed to 0.5 μM MEDS433 from −2 to −1 h prior to IAV infection (MOI of 0.1) (pre-treatment, Pre-T); during infection (adsorption stage, from −1 to 0 h; co-treatment, Co-T); after viral adsorption (from 0 to 48 h p.i.; post-treatment, Post-T); or during all phases (full treatment, Full-T). At 48 h p.i., the cell supernatants were harvested and titrated for IAV infectivity as described above.

To evaluate the effect of uridine, cytidine, DHO, or ORO addition, MDCK and A549 cells were seeded as describe above for the PRA and the VRA, and after 24 h, the cultures were infected with IAV (50 PFU/well or 0.001 PFU/cell) and treated with increasing concentrations of uridine, cytidine, ORO, or DHO in the presence of 0.4 μM of MEDS433. The drugs were maintained throughout the assay, and at 48 h p.i.: for the PRA, the cell monolayers were fixed and viral plaques microscopically counted; for the VRA cells, the supernatants were harvested and IAV yield titrated on MDCK cells.

To investigate the effect of blocking both the de novo biosynthesis and the salvage pathways of pyrimidines, A549 cell monolayers were infected with IAV (0.001 PFU/cell) and treated with 0.4 μM of MEDS433 and increasing concentrations of DPY in the presence of 20 μM uridine, which exceeds the physiological uridine plasma levels [33]. After 48 h p.i., the supernatants were harvested and IAV yield titrated on MDCK cells.

The combination of MEDS433 and DPY was examined by the VRA as describe above. Briefly, MEDS433 was added to A549 cells at 0.25×, 0.5×, 1×, 2×, and 4× EC_50_ alone or in combination with 3 μΜ DPY. The infection was performed with IAV (0.001 PFU/cell), and after 48 h p.i., the supernatants were collected and the influenza virions titrated on MDCK.

To assess the effects of the combination of MEDS433 and NHC on IAV replication, compounds, alone or in combination, were added to A549 cell monolayers at equipotent ratio of 0.25×, 0.5×, 1×, 2×, and 4× EC_50_ of each drug. Then, the cells were infected with IAV (0.001 PFU/cell), and after 48 h p.i., the supernatants were harvested and IAV yield titrated on MDCK cells.

The effect of the two-drug combinations was determined by means of the Chou method [35] as computed in the CompuSyn software 1.0 (http://www.combosyn.com (accessed 3 August 2022)) [36]. With this method, a combination index (CI) = 1 represents an additive effect, a CI value *>* 1 means antagonism, and a CI value *<* 1 indicates synergism.

### 2.5. Immunoblotting

Total cell protein extracts were prepared at different times p.i. from MDCK cell monolayers infected with IAV at an MOI of 0.1 PFU/cell and treated with 0.5 μΜ MEDS433 [37]. Subsequently, equal amounts of protein extracts were fractionated by 8% SDS-PAGE and transferred to PVDF membranes (Bio-Rad (Hercules, CA, USA). Filters were blocked for 2 h at 37 °C in 5% non-fat dry milk in 10 mM Tris-HCl (pH 7.5), 100 mM NaCl, and 0.05% Tween 20 and then immunostained with either the rabbit anti-IAV HA pAb (PA5-34929; Thermo Fisher Scientific (Waltham, MA, USA) (diluted 1:3000) or the rabbit anti-IAV NA pAb (PA5-32238; Thermo Fisher Scientific) (diluted 1:1500). The rabbit anti-GAPDH mAb (D16H11, Cell Signaling) (diluted 1:1000) or the mouse anti-vinculin mAb (V9264; Sigma-Aldrich) (diluted 1:4000) was used as the control for protein loading. Immunocomplexes were detected with a goat anti-rabbit Ig Ab conjugated to horseradish peroxidase (Life Technologies, Carlsbad, CA, USA) or with a goat anti-mouse Ig Ab conjugated to horseradish peroxidase (Life Technologies) and visualized by enhanced chemiluminescence (Western Blotting Luminol Reagent, Santa Cruz, CA, USA).

### 2.6. hDHODH Gene Silencing

For RNA interference, A549 cells were transfected with a *h*DHODH-targeting small interfering RNA (siRNA) (Origene-SR319917C) or a universal scrambled negative control siRNA (Origene-SR30004) using the siTran 2.0 siRNA Transfection Reagent (Origene, Rockville, MD, USA) and in agreement with the manufacturer’s protocol. Expression of the *h*DHODH protein was analyzed at 48 h post-transfection by immunoblotting using the mouse anti-*h*DHODH mAb (SC-166348, Santa Cruz Biotechnology, Santa Cruz, CA, USA) (diluted 1:200). To assess the effect of the *h*DHODH silencing on IAV replication, at 24 h post-transfection, A549 cells were infected with IAV at an MOI of 0.001 PFU/cell, and after 48 h p.i., the cell supernatants were harvested and titrated on MDCK cells.

### 2.7. Data Analysis

All data were generated from at least three independent experiments performed in triplicate. Statistical analysis was performed using GraphPad Prism version 8.0. Data are presented as the means ± SDs and considered to be statistically significant for *p* < 0.05.

## 3. Results

### 3.1. hDHODH Expression Is Required for Efficient IAV Replication

To evaluate the importance of *h*DHODH in the IV replication cycle, its expression was knocked down by siRNAs in the relevant cell model of human alveolar basal epithelial cells A549 (Figure 1A). As shown in Figure 1B, the reduction of *h*DHODH protein expression significantly affected IAV growth with a 183-fold reduction of the release of infectious virus particles at 48 h p.i., compared to both wild-type A549 cells and cells transfected with a scrambled negative siRNA control. These results therefore sustain the requirement of *h*DHODH for efficient IAV replication.

### 3.2. MEDS433 Inhibits IAV and IBV In Vitro Replication

Given the relevance of *h*DHODH for IAV replication, the pharmacological targeting of this enzymatic activity may be exploited to identify anti-IV molecules. Having this in mind, firstly, virus yield reduction assays (VRAs) were performed in A549 cells to investigate the possible anti-IV activity of the new *h*DHODH inhibitor MEDS433 (Figure 2A). As shown in Figure 2B,C, the measurement of the IV yield produced by A549 cells exposed to MEDS433 revealed a remarkable concentration-dependent inhibitory effect on both IAV and IBV replication. As for IAV replication, the EC_50_ and EC_90_ values were 0.064 ± 0.01 μM and 0.264 ± 0.002 μM, while for IBV, they were 0.065 ± 0.005 μΜ and 0.365 ± 0.09 μΜ, respectively (Table 1). It is noteworthy that MEDS433 was more effective than brequinar in inhibiting IV replication, since for this *h*DHODH inhibitor, an EC_50_ of 0.495 ± 0.027 μΜ and of 0.273 ± 0.014 μΜ was measured against IAV and IBV, respectively (Figure 2B,C).

Thereafter, given the HTA feature of MEDS433, a second human airway epithelial cell line was evaluated to investigate its anti-IV activity in a different host cell system. Hence, VRAs were performed in Calu-3 cells. In this cell line, MEDS433 produced EC_50_ and EC_90_ values of 0.055 ± 0.003 μΜ and 0.675 ± 0.05 μΜ against IAV and of 0.052 ± 0.006 μΜ and 0.807 ± 0.08 μΜ for IBV (Table 1).

To exclude that the anti-IV activity of MEDS433 might be due to the cytotoxicity of target cells, its effect on the viability of uninfected A549 and Calu-3 cells was evaluated by the MTT method, which produced a cytotoxic concentration 50 (CC_50_) for A549 cells of 64.25 ± 3.12 μΜ, with a favorable selectively index (SI) of 1104 for IAV and 988 for IBV, and a CC_50_ value of 54.67 ± 3.86 μΜ for Calu-3 cells, with an SI of 994 for IAV and 1051 for IBV (Table 1). Thus, the anti-IV activity of MEDS433 was not due to the inhibition of cell viability.

Lastly, to rule out the possibility that the antiviral effects of MEDS433 against the IVs was derived from the type of antiviral assay, plaque reduction assays (PRAs) were performed in MDCK cells infected with IAV or IBV. As depicted in Figure 2D,E, a MEDS433-mediated concentration-dependent inhibitory effect on both IAV and IBV replication in MDCK cells was again measured. As reported in Table 1, the EC_50_ and EC_90_ produced by MEDS433 in MDCK cells were 0.141 ± 0.021 μM and 0.256 ± 0.052 μΜ against IAV, while for IBV, they were 0.170 ± 0.019 μM and 0.330 ± 0.013 μΜ, respectively. Again, MEDS433 was more potent than brequinar against IAV and IBV even in MDCK cells, since the EC_50_ of the latter were 0.780 ± 0.012 μM against IAV and 1.07 ± 0.07 μM against IBV. Finally, the measurement of the cell viability of uninfected MDCK cells exposed to MEDS433 determined a CC_50_ of 119.8 ± 6.21 μΜ and an SI of 850 for IAV and 705 for IBV, thus confirming that even in canine cells, the anti-IV activity of MEDS433 was not due to the cytotoxicity of the target cells (Table 1).

Taken together, these results indicated that MEDS433 carried out a potent anti-IV activity that was independent of the type of IV, the cell line, or assay used.

### 3.3. MEDS433 Affects IAV Protein Expression by Targeting a Post-Entry Phase of the Virus Replicative Cycle

To investigate more in detail the anti-IV activity of MEDS433, cultures of IAV-infected MDCK cells were treated with MEDS433, and at various times p.i., total cell protein extracts were prepared and analyzed by immunoblotting for HA and NA protein content to monitor the levels of these representative IV proteins. As depicted in Figure 3, starting from 16 h p.i., the time point at which both IV glycoproteins became detectable, MEDS433 prevented their accumulation in infected and treated cells, thus indicating its ability to target a synthetic step in the virus replicative cycle.

Subsequently, time-of-addition experiments were carried out to pinpoint which phase of the IV replicative cycle was targeted by MEDS433. To this end, MDCK cells were treated with MEDS433 (0.5 μM) from −2 to −1 h prior to IAV infection (pretreatment (Pre-T)); or during IAV infection (from −1 to 0 h, adsorption stage; cotreatment (Co-T)); or after viral adsorption (from 0 to 48 h p.i.; post treatment (Post-T)); or from −2 to 48 h p.i. (full treatment (Full-T)). Infectious IAV particles released in cell supernatants were harvested at 48 h p.i. and titrated by the plaque assay. As shown in Figure 4A, MEDS433 was ineffective in interfering with the early phases of IAV’s replicative cycle. By contrast, it produced a severe reduction of the IAV titer of about three orders of magnitude when added at a post-entry stage (Post-T) or left on the cell from −2 to 48 h p.i. (Full-T), thus in agreement with its ability to block HA and NA protein accumulation at late times of infection (Figure 3).

Immunoblot analysis of protein extracts prepared from the very same IAV-infected and MEDS433-treated MDCK cells confirmed a severe impairment of HA and NA expression only by the Post-T and Full-T treatments, while the Pre-T and Co-T were ineffective (Figure 4B).

The results of this section therefore suggested that the antiviral activity of MEDS433 stems from an interference with a biosynthetic step essential for IV replication and that takes places at a post-entry stage. However, as the levels of viral mRNAs were not measured, the effect of MEDS433 on the expression of HA and NA proteins could be due to the inhibition of the synthesis of the corresponding mRNAs or their translation.

### 3.4. The Pyrimidine Biosynthesis Pathway in Implicated in the Anti-IV Activity of MEDS433

The results of Section 3.3 evoked a mechanism of the anti-IV activity of MEDS433 that is well suited to its ability to inhibit *h*DHODH activity, thus depleting the intracellular pyrimidine pool. To verify this hypothesis, we investigated whether the anti-IV effect of MEDS433 could be reversed by the addition of increasing concentrations of exogenous pyrimidine ribonucleosides, such as uridine or cytidine.

As shown in Figure 5A, IAV replication in MDCK cells was restored more than 50% by the addition of a 100-fold excess of both uridine and cytidine relative to the MEDS433 concentration used (0.4 μΜ). In human A549 cells, the effect of uridine addition was even more pronounced, since 50% of IAV replication was already achieved at a uridine concentration 50-fold higher relative to MEDS433, while it was completely brought back to the level of untreated control by concentrations of both uridine and cytidine that were 1000-/2000-fold higher (Figure 5C). These results clearly indicated that the de novo pyrimidine synthesis pathway was targeted by MEDS433 in IAV-infected cells.

To further confirm that the inhibition of *h*DHODH enzymatic activity by MEDS433 was effectively responsible for its antiviral activity, IAV-infected MDCK cells were treated with MEDS433 in the presence of increasing concentrations of dihydroorotic acid (DHO), or orotic acid (ORO), inasmuch as they are the *h*DHODH substrate or its product, respectively. Figure 5B shows that ORO significantly reversed the inhibitory effect of MEDS433 already from a concentration 100-fold that of MEDS433. Conversely, the addition of DHO, even at 800 μΜ (2000-times more than the MEDS433 concentration used), did not reduce the anti-IAV activity of the *h*DHODH inhibitor (Figure 5B), thus sustaining that, indeed, MEDS433 inhibits a step of the pyrimidine biosynthesis pathway that is downstream from DHO. Similar results were obtained also in A549 cells (Figure 5D).

Together, these results indicated that MEDS433 inhibits IV replication through a specific inhibition of the *h*DHODH enzymatic activity in IV-infected cells, therefore blocking the oxidation of DHO to ORO in the de novo synthesis of pyrimidines.

### 3.5. The Combination of MEDS433 with an Inhibitor of the Nucleoside Salvage Pathway Enhances the Anti-IAV Activity of the hDHODH Inhibitor

Given that uridine counteracted the anti-IAV activity of MEDS433 (Figure 5), it could be possible that its physiological plasma concentration may limit the antiviral strength of MEDS433 in the host through the salvage pathway.

To address this problem, first, we investigated the effects on IAV replication of the combination of MEDS433 with dipyridamole (DPY), an inhibitor of the nucleoside/nucleotide transport channel hENT1/2 implicated in the pyrimidine salvage pathway [38,39,40]. For this purpose, VRAs were performed in IAV-infected A549 cells to determine the effect of the combination of a 0.25-, 0.5-, 1-, 2-, or 4-fold of MEDS433 EC_50_ to 3 µM DPY ratio. It is noteworthy that DPY as a single agent did not exert any inhibitory activity on IAV replication up to 10 µM (Figure 6). Nevertheless, when 3 µM DPY was used in combination with the different concentrations of MEDS433, it increased the anti-IAV potency of the *h*DHODH inhibitor (Figure 6); in fact, the EC_50_ of MEDS433 (0.063 ± 0.044 μM) was reduced to 0.011 ± 0.001 μM by the combination with DPY. The computed combination index (CI) values [31,32] corroborated that the combination of MEDS433 with DPY resulted in a synergistic antiviral activity at any of the MEDS433′s concentrations tested, since all the CIs were *<*0.9 (Table 2).

Thereafter, we tested the efficacy of the MEDS433-DPY synergistic combination even in the presence of a hyper-physiological concentration of uridine (20 μM), far exceeding its plasma concentrations [41]. As shown in Figure 7, the presence of exogenous uridine reversed, as expected, the inhibitory activity of a MEDS433 concentration (0.4 μΜ) that remarkable inhibited IAV replication when tested as a single agent (Figure 2B). Conversely, the addition of increasing amounts of DPY restored the antiviral activity of MEDS433, thus sustaining the applicability of this combination to inhibit IAV replication, even in the presence of exogenous uridine (Figure 5A). Furthermore, none of the tested combinations reduced the viability of A549 cells, thus confirming that the DPY-mediated restoration of the MEDS433′s anti-IV activity was not due to a nonspecific cytotoxic effect.

Taken together, the results of this section suggested that a combination of two modulators of the pyrimidine metabolism, such as a *h*DHODH inhibitor and an inhibitor of the salvage pathway, was effective against IV replication, even in the presence of uridine concentrations that exceed in vivo host conditions.

### 3.6. MEDS433 and the Ribonucleoside Analogue N^4^-Hydroxycytidine Synergistically Act against IAV Replication

Lastly, we investigated whether the combination of MEDS433 with a DAA targeting IV RNA replication could result in a synergistic, additive, or antagonistic effect against IV replication. To this end, first, we measured the EC_50_ of N^4^-hydroxycytidine (NHC or EIDD1931), a cytosine analogue that exerts a potent anti-IV antiviral activity as a result of increased viral mutagenesis following its incorporation by IV RdRp [42,43]. The EC_50_ of NHC against IAV, as measured by VRA in A549 cells, was 0.332 ± 0.011 μM. Then, VRAs were performed with different concentrations of both MEDS433 and NHC corresponding to 0.25-, 0.5-, 1-, 2-, or 4-fold their EC_50_ values to obtain equipotent ratios (MEDS433 EC_50_/NHC EC_50_). As depicted in Figure 8, the anti-IAV efficacy of NHC was increased by the combination with MEDS433, since the EC_50_ of NHC was decreased to 0.124 ± 0.011 μM by the combination with the *h*DHODH inhibitor. The synergism between MEDS433 and NHC was confirmed by the computed CI values *<* 0.9 at any of the MEDS433 to NHC combinations tested (Table 3). These results therefore suggested that a combination of MEDS433 with a RdRp-targeting DAA, such as NHC, might be of interest to design new pharmacological approaches for the control of IV infections.

## 4. Discussion

The ongoing COVID-19 pandemic, along with the frequent emergence and re-emergence of human respiratory viruses in recent decades have renewed interest in host-targeting antivirals. Since HTAs target cellular biochemical pathways or factors that can be exploited by many different viruses for their replication, they may exert broad-spectrum antiviral activity and, thus, represent valuable tools in the preparedness against future viral infections [44]. Given the essential role of *h*DHODH in the de novo pyrimidine biosynthesis pathway and its indispensable requirement for efficient viral replication, as we strengthened here for IVs (Figure 1), this host enzymatic activity is considered a reliable HTA target to develop new broad-spectrum antivirals, also against antigenically different IVs [45,46]. To further this prospect sustain, this study reported that the new *h*DHODH inhibitor MEDS433 carries out a potent dose-dependent antiviral activity against IAV and IBV, with a mechanism that derives from a selective inhibition of *h*DHODH activity in IV-infected cells. Our findings corroborate the feasibility of the pharmacological targeting of *h*DHODH to control IV replication, as recently observed also for other *h*DHODH inhibitors, such as brequinar [28], FA-613 [47], and S312 and S416 [48].

MEDS433 is a small molecule belonging to a novel class of *h*DHODH inhibitors that are based on a 2-hydroxypyrazolo [1,5-a]-pyridine scaffold [24,25,26]. It was designed by applying a bioisosteric approach to brequinar, one of the most-potent *h*DHODH inhibitors [27]. Even though both MEDS433 and brequinar target with high affinity the ubiquinone binding site of *h*DHODH and show a similar potency in inhibiting its enzymatic activity [24], when they were compared for anti-IV activity, MEDS433 resulted in being more effective (Figure 2). The superior activity of MEDS433 against IV replication might depend on its more favorable lipophilicity (logD7.4 value of 2.35) in comparison to the more polar brequinar (logD7.4 value of 1.83) [26]. A value of logD7.4 equal to 2.50 has been suggested as the optimum for the inhibition of mitochondrial *h*DHODH, since inhibitors with a lower logD7.4 may poorly cross the mitochondrial membrane, while those endowed with a higher logD7.4 may display a reduced cellular adsorption [49]. Furthermore, we observed that MEDS433 was less cytotoxic than brequinar; in fact, the CC_50_ measured in A549 cells of brequinar (5.69 μΜ) was about ten-fold lower than that of MEDS433 (64.25 μΜ), which determined SI values of 12 for IAV and 21 for IBV, respectively. Again, the optimal lipophilicity of MEDS433 might reduce the possibility of off-targeting effects within host cells, thus explaining the high SI values measured for this *h*DHODH inhibitor. The low cytotoxicity of MEDS433 is undoubtedly a favorable feature and, indeed, contributes to making it a promising HTA candidate.

However, to become a candidate for the treatment of respiratory virus infections, as IVs, an optimal HTA, in addition to being potent and having a low cytotoxicity, as we observed for MEDS433, it should be suitable even for combination treatments to enhance antiviral efficacy and to reduce the risk of the emergence of drug resistance [44]. Keeping this in mind, it is worth noting that this study adds a couple of new pieces of knowledge that may be of interest to be considered toward the validation of *h*DHODH inhibitors to the design of new therapeutic approaches for the management of IV infections.

The first is related to the limited therapeutic efficacy that has been reported for some *h*DHODH inhibitors in animal models of human viral infections, despite their clear in vitro antiviral activity [50,51,52]. This failure likely derives from the inadequate antiviral potency and pharmacokinetics performance of the evaluated *h*DHODH inhibitors, as well as from the counteracting effect of the pyrimidine salvage pathway by means of the transport of exogenous uridine into virus-infected cells. Accordingly, the pharmacological targeting of the pyrimidine salvage pathway may be beneficial, inasmuch as it may enhance the in vivo antiviral effect of *h*DHODH inhibitors.

Relevant to this hypothesis is the observations that a combination of MEDS433 with DPY was synergistic against IAV replication (Figure 6), even in the presence of amounts of uridine that exceed the physiological plasma concentration (Figure 7). In fact, notwithstanding that DPY is devoid of anti-IV activity (Figure 6), when used in combination with the *h*DHODH inhibitor, it restored successfully the antiviral activity of a MEDS433 concentration no longer effective as a consequence of uridine supplementation (Figure 7). DPY is a pyrimidopyrimidine derivative that inhibits the equilibrative nucleoside transporters (ENT) 1 and 2, the most effective nucleoside/nucleotide transporters of the pyrimidine salvage pathway [40]. Thanks to its ability to block the uptake of adenosine into platelets, endothelial cells, and erythrocytes, DPY hinders platelet aggregation and vasodilatation. Therefore, DPY is approved as an oral agent in the prophylaxis of thromboembolism in cardiovascular disease [39]. While the combination of DPY with *h*DHODH inhibitors has been proposed to increase the anticancer effects of the latter [53], the potential of this combination against viral infections has been investigated poorly. In this regard, recently, we observed that the combination of MEDS433 with DPY was effective also against HSV-1 and SARS-CoV-2, even in the presence of a hyperphysiological concentration of uridine [30,31]. Importantly, the concentration of DPY (3 µM) that we observed to be synergistic in combination with MEDS433 against IAV replication is lower than the DPY Cmax (2.2 μg/mL, which corresponds to 4.4 μM) [54] and, thus, clinically achievable in patients undergoing DPY therapy. Thus, the wide clinical experience of DPY could allow its rapid repositioning against virus infections, including IVs, to be clinically useful in combination with *h*DHODH inhibitors.

Furthermore, pertinent to the possibility to exploit combinations between compounds targeting the pyrimidine synthesis pathway and DAA against IVs is the observation that several combinations of MEDS433 and N4-hydroxycytidine (NHC) interact in a synergistic manner, each reinforcing the other’s antiviral activity against IAV (Figure 8). To our knowledge, this is the first observation of a synergistic effect between NHC and a *h*DHODH inhibitor against IVs.

The ribonucleoside analog NHC is the active metabolite of the prodrug molnupiravir, which was initially developed for IV infections [42,43], then repurposed against SARS-CoV-2, currently approved for use in COVID-19 [55]. NHC, once phosphorylated to the pharmacologically active NHC-triphosphate (NHCTP), is incorporated into nascent RNA by viral RdRps. However, the tautomeric interconversion within NHC then leads to misincorporation of adenine bases instead of guanine bases in the subsequent cycles of viral RNA synthesis, which eventually results in viral mutagenesis, inhibition of viral replication, and antiviral activity [56].

Recently, the combination of NHC with different *h*DHODH inhibitors has been reported to exert a synergistic effect against in vitro replication of SARS-CoV-2 and in reducing viral titers and the pathology in animal models of the infection [57,58]. Since NHCTP competes with cellular CTP for incorporation into nascent RNA, it has been suggested that the impairment of CTP synthesis by *h*DHODH inhibitors reduces this competition and facilitates the incorporation of NHCTP into newly synthetized SARS-CoV-2 RNA. Thereby, *h*DHODH inhibitors potentiate the antiviral effectiveness of NHC. This ribonucleoside analog is in fact imported within cells by the salvage pathway, and therefore, the intracellular NHCTP levels are not affected by the *h*DHODH inhibition [57,58].

Since combinations of different antivirals have proven to be effective in the control of many viral infections, the combination of broad-spectrum nucleoside analogues, such as NHC, with DHODH inhibitors might be suitable to develop therapeutic strategies to control infections of a wide range of viruses.

Lastly, an important added value of MEDS433 is its antiviral effectiveness against a broad range of viruses that cause respiratory tract infections in humans, since, in addition to IAV and IBV, we observed the ability of MEDS433 to prevent the replication of hCOV-229E, hCoV-OC43, and SARS-CoV-2 [31]. This feature and its inherent HTA advantage of the low risk of emergence of drug-resistant strains make MEDS433 of interest for the development of broad-spectrum antiviral agents, even in the preparedness against future emerging human respiratory viruses given the independence of its antiviral effects with respect to a specific virus. The disastrous consequences of COVID-19 are indeed indisputable evidence of the need for new broad-spectrum antiviral drugs known to be effective against respiratory viruses that may emerge from future zoonoses.

## 5. Conclusions

In conclusion, this study indicated MEDS433 as an attractive novel HTA candidate endowed with advantageous features, such as a potent antiviral activity against both IAV and IBV and rapidly deployable against future novel emerging IVs. Moreover, MEDS433 could also be considered for combination drug treatments with both nucleoside analogues and other anti-pyrimidines, such as NHC and DPY, for the design of new therapeutic approaches to IV infections. Presently, the potent in vitro anti-IV activity of MEDS433 and its valuable drug-like profile support further studies to evaluate its efficacy in preclinical animal models of IV infections.

## Figures and Tables

**Figure 1 viruses-14-02281-f001:**
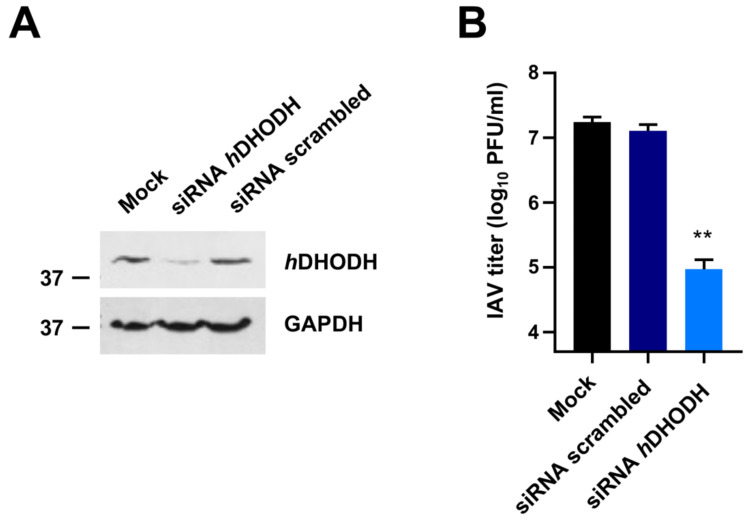
Knockdown of *h*DHODH affects IAV replication. (**A**) Expression of *h*DHODH in A549 cells transfected with *h*DHODH or scrambled negative control siRNAs. At 48 h after of transfection, total protein extracts were prepared, fractionated by 10% SDS-PAGE, and analyzed by immunoblotting with anti-*h*DHODH mAbs. Immunodetection of GAPDH was used as a control for protein loading. Molecular weight markers are shown beside the left side of each panel. (**B**) VRA was performed in A549 cells mock-transfected or transfected with *h*DHODH or scrambled siRNAs. At 24 h after transfection, cells were infected with IAV, and after 48 h p.i., the cell supernatants were harvested and titrated by the plaque assay in MDCK cells. The data shown are the means ± SDs of two independent experiments performed in triplicate and analyzed by the unpaired *t*-test, corrected at FDR q < 0.05. ** (*p* < 0.001) compared to the calibrator sample (Mock).

**Figure 2 viruses-14-02281-f002:**
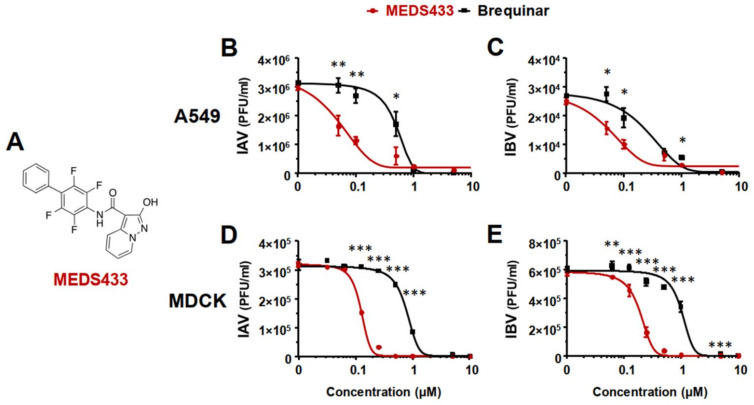
Inhibition of IAV and IBV replication by MEDS433. (**A**) Chemical structure of MEDS433, the *h*DHODH inhibitor investigated as an anti-IV agent in this study. (**B**,**C**) VRAs were performed in A549 cells infected with IAV (**B**) or IBV (**C**) and treated with increasing concentrations of MEDS433 or brequinar 1 h before, during, and post-infection. At 48 h p.i., the cell supernatants were harvested and titrated by the plaque assay in MDCK cells. (**D**,**E**) PRAs were performed in MDCK cell monolayers infected with IAV (**D**) or IBV (**E**) and, where indicated, treated with increasing concentrations of MEDS433 or brequinar. At 48 h p.i., the viral plaques were microscopically counted and converted in viral titer (PFU/mL). The concentrations of MEDS433 or brequinar producing 50% and 90% reductions in plaque formation (EC_50_ and EC_90_) were then calculated. Data shown are the means ± SDs (error bars) of three independent experiments performed in triplicate and analyzed by a two-way ANOVA, followed by Dunnett’s multiple comparison test. Statistical analysis was performed by comparing MEDS433-treated samples with the brequinar-treated samples for each condition. * (*p* < 0.01); ** (*p* < 0.001); *** (*p* < 0.0001).

**Figure 3 viruses-14-02281-f003:**
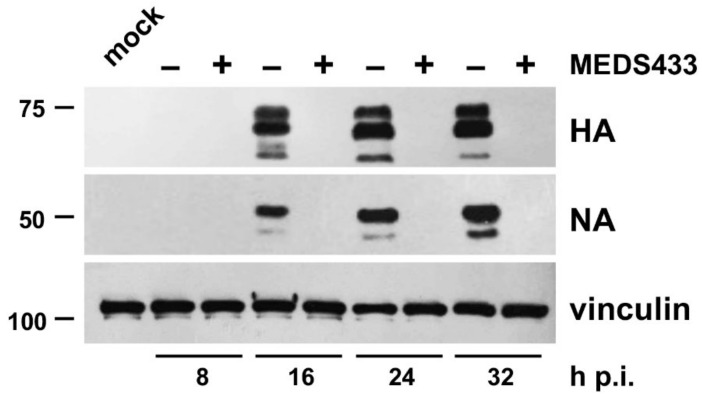
MEDS433 impairs the expression of IAV HA and NA proteins. MDCK cells were infected with IAV at an MOI of 0.1 PFU/cell and, where indicated, treated with 0.5 μΜ MEDS433 or DMSO as a control. Total cell protein extracts were prepared at the indicated times p.i., fractionated by 8% SDS-PAGE, and analyzed by immunoblotting with anti-IAV HA and anti-IAV NA pAbs. Vinculin immunodetection was used as a control for protein loading. Molecular weight markers are shown at the left side of each panel.

**Figure 4 viruses-14-02281-f004:**
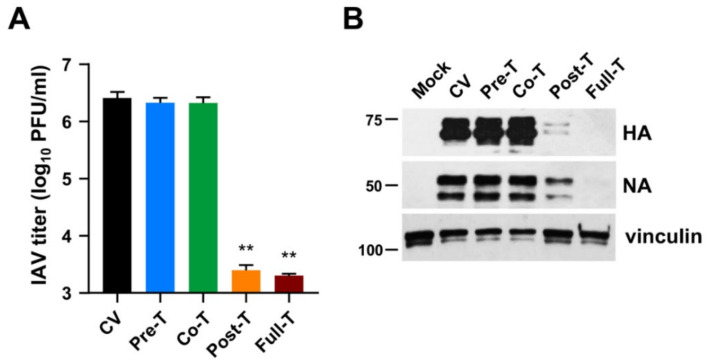
MEDS433 targets a post-entry phase of the IAV replicative cycle. MDCK cell monolayers were infected with IAV at an MOI of 0.1 and, where indicated, treated with 0.5 μΜ MEDS433 1 h prior to the infection (from −2 to −1 h, **Pre-T**), or during the infection (from −1 to 0 h, **Co-T**), or after infection (from 0 to 48 h p.i., **Post-T**;), or from −2 to 48 h p.i. (**Full-T**). Mock-infected cells (**Mock**) and control IAV-infected cells (**CV**) were exposed to DMSO only. (**A**) Cell supernatants harvested at 48 h p.i. were titrated by the plaque assay, and viral plaques were microscopically counted and plotted as PFU/mL. The data shown are the means ± SD of two independent experiments performed in triplicate and analyzed by a one-way ANOVA followed by Dunnett’s multiple comparison test. ** (*p* < 0.001) compared to the calibrator sample (**CV**). (**B**) Total cell extracts were prepared at 48 h p.i., fractionated by 8% SDS-PAGE, and analyzed by immunoblotting with anti-IAV HA and anti-IAV NA pAbs. Vinculin immunodetection was used as a control for protein loading. Molecular weight markers are shown next to the left side of each panel.

**Figure 5 viruses-14-02281-f005:**
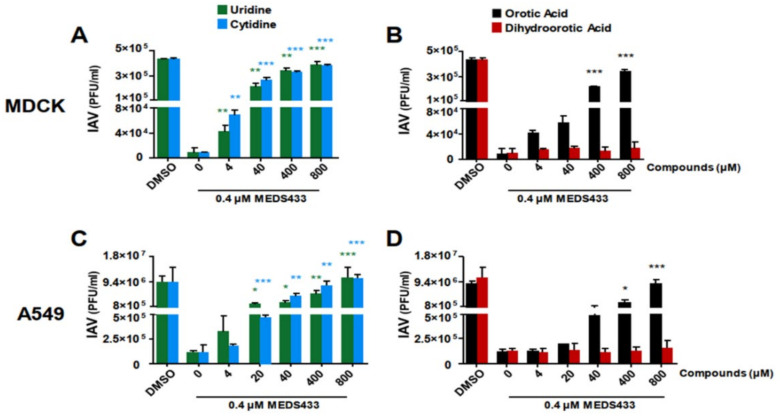
Uridine and orotic acid reversed the inhibitory effect of MEDS433 on IAV replication. (**A**,**B**) PRAs were performed in IAV-infected MDCK cells treated before, during, and post-infection with 0.4 μΜ MEDS433 in the presence of increasing concentrations of uridine or cytidine (**A**) and dihydroorotic acid or orotic acid (**B**). At 48 h p.i., the viral plaques were stained and microscopically counted. (**C**,**D**) VRAs were performed in A549 cells infected with IAV and treated with 0.4 μΜ MEDS433 in the presence of increasing concentrations of uridine (**C**) and dihydroorotic acid or orotic acid (**D**). At 48 h p.i., the cell supernatants were harvested and titrated in MDCK cells. Data shown represent means ± SDs of three independent experiments performed in triplicate and analyzed by a one-way ANOVA, followed by Dunnett’s multiple comparison test. * (*p* < 0.01); ** (*p* < 0.001); *** (*p* < 0.0001) compared to the calibrator sample (MEDS433 alone).

**Figure 6 viruses-14-02281-f006:**
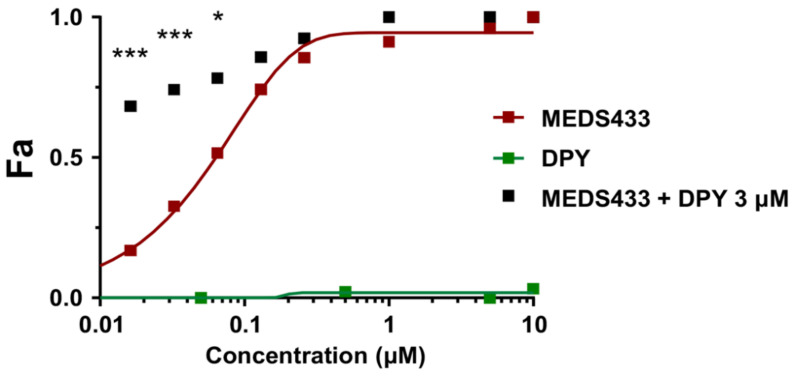
Effects of the combination of MEDS433 and dipyridamole on IAV replication. VRAs were performed in A549 cells treated with MEDS433 alone or in combination with different concentrations of DPY. At 48 h p.i., the cell supernatants were harvested and titrated by the plaque assay in MDCK cells. The effect of the combination was then analyzed by the CompuSyn software [36] and displayed as a fractional effect analysis (Fa) plot in relation to the compound concentrations. The anti-IAV activity of MEDS433 and DPY when used as single agents is depicted by red and green Fa curves, respectively. The effect of the MEDS433-DPY combination is shown with black squares. Results are representative of three independent experiments performed in triplicate and analyzed by a two-way ANOVA, followed by Dunnett’s multiple comparison test. Statistical analysis was performed by comparing MEDS433-treated samples with the MEDS433 + DPY-treated samples for each condition. * (*p* < 0.01); *** (*p* < 0.0001).

**Figure 7 viruses-14-02281-f007:**
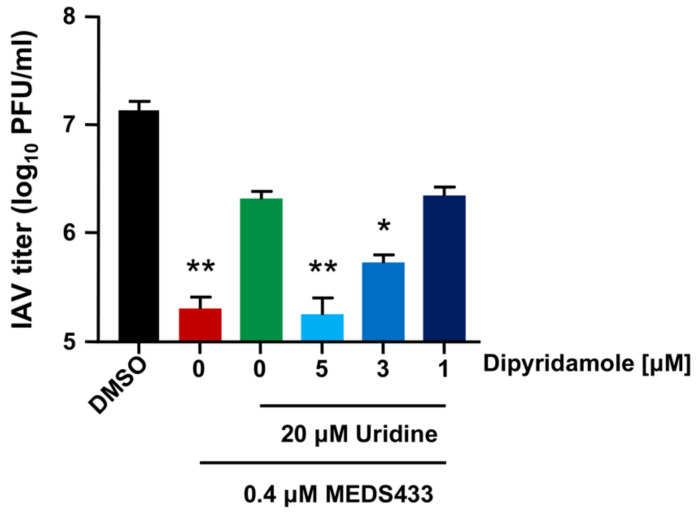
A combination of modulators of pyrimidine metabolism inhibits IAV replication in the presence of exogenous uridine. VRAs were performed in A549 cells treated with MEDS433 alone or in combination with different concentrations of DPY. Where indicated, infected cells were supplemented with 20 μM uridine. At 48 h p.i., the cell supernatants were harvested and titrated by the plaque assay in MDCK cells. The data shown represent means ± SDs of three independent experiments performed in triplicate, and MEDS433-treated samples were analyzed by an unpaired *t*-test, corrected at FDR q < 0.05. * (*p* < 0.01); ** (*p* < 0.001) compared to the calibrator sample (MEDS433 + 20 μM uridine, green column).

**Figure 8 viruses-14-02281-f008:**
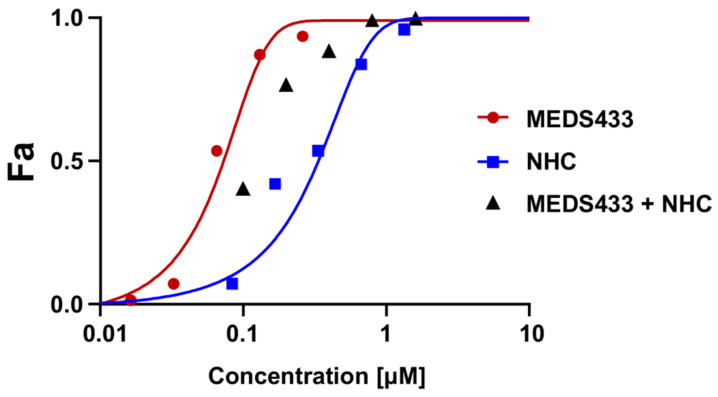
The combination of MEDS433 with NHC is synergistic against IAV replication. A549 cells were treated with different concentrations of MEDS433 alone or in combination with various concentrations of NHC before and during IAV infection. At 48 h p.i., the cell supernatants were harvested and titrated for IAV by the plaque assay in MDCK cells. Viral titers were analyzed by the CompuSyn software 1.0 [36], and the anti-IAV activity is shown as a fractional effect analysis (Fa) plot in relation to the drugs’ concentrations. Red and blue Fa curves represent the activity of MEDS433 or NHC on IAV replication when employed as single agents, respectively. The effect of the MEDS433–NHC combination is shown with black triangles. Results are representative of three independent experiments performed in triplicate.

**Table 1 viruses-14-02281-t001:** Antiviral activity of MEDS433 against representative influenza viruses.

IV	Cell Line	EC_50_ (μM) ^a^	EC_90_ (μM) ^b^	CC_50_ (μM) ^c^	SI ^d^
IAV	A549	0.064 ± 0.01 μM	0.264 ± 0.002 μM	64.25 ± 3.12 μΜ	1104
	Calu-3	0.055 ± 0.003 μΜ	0.675 ± 0.05 μΜ	54.67 ± 3.86 μΜ	994
	MDCK	0.141 ± 0.021 μM	0.256 ± 0.052 μΜ	119.8 ± 6.21 μΜ	850
IBV	A549	0.065 ± 0.005 μΜ	0.365 ± 0.09 μΜ	64.25 ± 3.12 μΜ	988
	Calu-3	0.052 ± 0.006 μΜ	0.807 ± 0.08 μΜ	54.67 ± 3.86 μΜ	1051
	MDCK	0.170 ± 0.019 μM	0.330 ± 0.013 μΜ	119.8 ± 6.21 μΜ	705

^a^ EC_50_, compound concentration that inhibits 50% of replication, as determined by the VRAs in A549 and Calu-3 cells, or by the PRAs in MDCK cells. ^b^ EC_90_, compound concentration that inhibits 90% of viral replication. ^c^ CC_50_, compound concentration that produces 50% cytotoxicity, as determined by the cell viability assays in A549, Calu-3, or MDCK cells. Reported values represent the means ± SDs of data derived from three experiments in triplicate. ^d^ SI, selectivity index (determined as the ratio between CC_50_ and EC_50_).

**Table 2 viruses-14-02281-t002:** Analysis of the effects of the combination of MEDS433 and DPY against IAV replication.

MEDS433 Concentration (Fold of EC_50_ ^a^) + DPY 3 μM	MEDS433/DPY CI ^b^	Drug Combination Effect ^c^ of MEDS433 and DPY
4×	0.824 ± 0.023	Moderate Synergism
2×	0.607 ± 0.116	Synergism
1×	0.341 ± 0.025	Synergism
0.5×	0.203 ± 0.044	Strong Synergism
0.25×	0.126 ± 0.032	Strong Synergism

^a^ The EC_50_ values of MEDS433 were determined by VRAs in IAV-infected A549 cells, as described in the Materials and Methods. ^b^ Combination index (CI), obtained by computational analysis with the CompuSyn software 1.0. Reported values represent means ± SDs of data derived from *n* = 3 independent experiments in triplicate. ^c^ According to the method of Chou [31], drug combination effects are defined as: strong synergism for 0.1 < CI < 0.3; synergism for 0.3 < CI < 0.7; moderate synergism for 0.7 < CI < 0.85; slight synergism for 0.85 < CI < 0.90; nearly additive for 0.90 < CI < 1.10; slight antagonism for 1.10 < CI < 1.20; moderate antagonism for 1.20 < CI < 1.45.

**Table 3 viruses-14-02281-t003:** Analysis of the effect of the combination of MEDS433 and NHC on IAV replication.

MEDS433/NHC Combination at Equipotent Ratio (fold of EC_50_ ^a^)	MEDS433/NHC CI ^b^	Drug Combination Effect ^c^ of MEDS433 and NHC
4×	0.075 ± 0.004	Very Strong Synergism
2×	0.484 ± 0.011	Synergism
1×	0.825 ± 0.004	Moderate Synergism
0.5×	0.604 ± 0.028	Synergism
0.25×	0.625 ± 0.019	Synergism

^a^ Fold of EC_50_ MEDS433/EC_50_ NHC yielding an equipotent concentration ratio (approximately 1:5.12) between the two combined drugs. The EC_50_ values were determined by VRAs against IAV, as described in the Materials and Methods. ^b^ Combination index (CI), obtained by computational analysis with the CompuSyn software 1.0. Reported values represent means ± SDs of data derived from *n* = 3 independent experiments in triplicate. ^c^ Drug combination effects are defined as: very strong synergism for CI < 0.1; strong synergism for 0.1 < CI < 0.3; synergism for 0.3 < CI < 0.7; moderate synergism for 0.7 < CI < 0.85; slight synergism for 0.85 < CI < 0.90; nearly additive for 0.90 < CI < 1.10; slight antagonism for 1.10 < CI < 1.20; moderate antagonism for 1.20 < CI < 1.45 [31].

## Data Availability

Not applicable.

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
