# Peer review of "The Novel hDHODH Inhibitor MEDS433 Prevents Influenza Virus Replication by Blocking Pyrimidine Biosynthesis"

_viruses, 2022, doi:10.3390/v14102281_

Round 1

Reviewer 1 Report

This submission describes MEDS433, a small molecule inhibitor of DHODH, carry out anti-IV activity. Using plaque assay and plaque reduction assay, the data presented demonstrated that the antiviral activity of MEDS433 is independent of virus strain and cell line and may takes places at late stage of replication. Additional results suggest that the MEDS433 and the ribonucleoside analogue synergistically inhibit the IV replication. The manuscript is well-written, and interpretations are supported by the data presented. The conclusions reached are justified. My concerns are few and highlighted below. 
1. Molecular weight markers are not included in Figure 1A.

2. Knockdown DHODH significantly affected IAV replication, how about KO?
3. 
The authors should provide the biological replicates for Figure 3.

4. Please check throughout for the format of citation.

Reviewer 2 Report

Influenza virus (IV) remains a major global health concern, and is listed as one of the viruses with the highest potential to emerge as a pandemic pathogen. There is intense interest in developing novel antiviral compounds which could be used to disrupt IV replication pathways. In addition, the targeting of host pathways is considered a major approach with potential to prevent resistance to drugs. Thus, the potential significance of the work is high in terms of forming the basis for new antiviral therapies.  

Strengths of the work include the importance of developing new anti-IV therapies, the targeting here of alternative pyrimidine pathways that are upregulated by IV infection, the direct assays for virus production (e.g., plaques) and the high-quality data. The data supporting the combined use of MED and DPY is interesting and potentially useful. The manuscript is very well written.  

A main weakness of the work is that the effect of the drug on virus yield (e.g. Fig 2, Fig. 7) is only about 3-fold or in some cases not much more than 1 log. While this may be significant statistically, it is not an impressive difference. Similarly, it is not clear how a small difference in plaque reduction correlates with the huge difference in appearance of viral glycoproteins by western blot. Is there any glycoprotein produced at all? The authors should clarify the relationship between plaque reduction and expression of viral glycoproteins which do not correlate. This could include analysis of 10x as much cell lysate for the drug treated samples compared to controls – this would show the relative effect on accumulation of glycoproteins.

Other comments:

1)    The title of the manuscript is very generic. It would be more helpful to the reader to be more specific about the target for the cmpd.

2)    The data in Fig 2 are interesting, but it is not clear what the titer of virus would be in the absence of drug…..this control appears to be missing.

3)    The data in Fig 2 show error bars but no statistical analysis. Is the difference between MED and Brequinar statistically significant? Without the control of no drug it is difficult to see how this could be calculated.

4)    The assays for effects of drug on virus use different MOIs – some use 50 PFU per well (what moi is this?), others MOI of 0.1 and others MOI of 0.001. This is confusing and does not paint a consistent picture of the mechanism.

5)    The data in Fig. 5 are interesting, but the graphs should be replotted to show the actual numbers for the low values. This way the reader can see the differences, which are masked by the small window on the y-axis for plotting.

6)    Fig. 6 lacks statistical analysis.

7)    The discussion lacks any mention of mechanism. How is the drug disrupting expression of viral glycoproteins?

8)    The discussion mentions that the authors have previously shown similar results with 229E, OC43 and SARS and RSV. It is important that this should be moved to the introduction, and used to state that the authors are now extending these prior results to IV. Thus, the reader will now know that the novelty of the application of MED to IV is tempered by the prior studies on other viruses.

Reviewer 3 Report

In this manuscript, the authors show the anti-influenza virus activity of MEDS433, a hDHODH inhibitor and synergistic effect of MEDS433 with DPY and NHC on anti-influenza virus activity. The development of host-targeted antivirals will become increasingly important, and the de novo pyrimidine biosynthesis pathway inhibited by MEDS433 is one potential target for antiviral drugs. While I agree with the authors' arguments in broad outlines, there are some concerns. I believe that improving these concerns would strengthen the authors' arguments.

My comments are as follows,

Overall,

The authors have shown synergistic effects of MEDS433 with DPY, an inhibitor of the pyrimidine salvage pathway, and NHC, a nucleoside analogue. One concern is that the lack of results for compounds that work competitively with MEDS433 makes it difficult to assess their synergistic effects on these compounds. In other words, if the results show that MEDS433 works competitively with other hDHODH inhibitors, then together with other results, a synergistic effect of hDHODH inhibition and inhibition of other pathways can be demonstrated.

L40-43

The reference describes the 2010-2011 season, but it would be better to describe the latest situation. Perhaps the percentage of oseltamivir-resistant viruses has not increased as much.

L46-48

Favipiravir resistant viruses should not have been isolated from clinical trials.

L66-68

The reference shows upregulation of the pyrimidine biosynthesis pathway in cytomegalovirus infection and elevated fatty acid synthesis in cytomegalovirus infection and influenza virus infection. Has upregulation of the pyrimidine biosynthesis pathway in influenza virus infection been shown?

Table 1

In general, studies of the anti-influenza virus activity of compounds are evaluated for activity with a variety of strains, including clinical isolates. Because the compounds used in this paper act on the host protein, it may not be necessary to evaluate activity using a variety of strains. However, it is necessary to evaluate anti-influenza virus activity using cell lines other than MDCK and A549 (for example: calu-3, NHBE, etc).

Figure 5

Is it not rescued by cytidine?

Figures 6 and 8

How did you calculate Fa? To my knowledge, Fa is calculated as (1-inhibition rate), and the higher the concentration of the compound, the lower the Fa.

It would be easier to understand if figures 6 and 8 used the same description of units for the concentration of compounds.

Figure 7

Isn't a multiple comparison test required?

Reviewer 4 Report

In this manuscript, these authors characterized the anti-IV activity of MEDS433, a novel small molecule inhibitor of the human dihydroorotate dehydrogenase (hDHODH). MEDS433 exhibited a potent antiviral activity against IVA and IVB replication that was reversed by the addition of exogenous uridine or the hDHODH product orotate, thus indicating that MEDS433 targets notably hDHODH activity in IV-infected cells. When MEDS433 was used in combination either with dipyridamole (DPY), an inhibitor of the pyrimidine salvage pathway, or with an anti-IV DAA, such as the N4-hydroxycytidine (NHC), synergistic anti-IV activities were observed. It is an interesting and meaningful research. The data presented are generally strong, and appear convincing. The work would benefit with further experiments to help strengthen the main conclusions and to better understand.

Major comments:

1.       Knockdown of DHODH in A549 cells in figure 1, what’s the target sequence? The author should choose at least 2 siRNA targeting the gene to conform the knockdown.

2.       The authors want to confirmed that MEDS433 inhibits virus by targeting hDHODH activity, they should confirmed that MEDS433 has low or have no anti-viral function in hDHODH knockdown or knockout cell lines. If they want to claim that MEDS422 affects the activity of hDHODH directly, maybe they also need to performed the interaction experiment between the drug and protein, even biochemical enzyme activity test in vitro.

3.       It would be perfect if the authors could provide more data of this drug, such as  pharmacokinetics, pharmacodynamics, even animal data.

Round 2

Reviewer 2 Report

This revised manuscript by Sibille et al, as addressed most of the comments from this reviewer. There are still some points that must be addressed and modified Figures to include, as they are not completely clear to a reader.

1)    “A main weakness of the work is that the effect of the drug on virus yield (e.g. Fig 2, Fig. 7) is only about 3-fold or in some cases not much more than 1 log.”

The authors provide a new Fig 2 (which they call Fig. R1 for some reason), which shows statistical significance. Despite the author’s view that this new figure is “already dense” this must be shown to convince a reader that there are significant differences. The figure is not already dense, and this makes it a much more convincing figure. Thank you for including.

As to the differences in infectious virus versus glycoprotein expression, the authors respond back that the mechanism of action of the drug is a post-entry step, likely involving inhibition of RNA synthesis. While the post-entry step is clearly shown in the data in Figure 4 (and is well done), there is an assumption that this is at the level viral RNA which was not measured.

Thus, the text should add the caveat that viral RNA was not measured, and thus, the effect could be on translation or production of viral mRNA.

2)     The original Fig 6 did not include statistical analysis.

The authors now provide a new Fig. R2 with this analysis but claim not to want to use this version. Please include the new version with the statistics to improve the analysis and rigor of the study.

Other comments from this reviewer were adequately addressed.

This revised manuscript by Sibille et al, as addressed most of the comments from this reviewer. There are still some points that must be addressed, as they are not completely clear to a reader.

1)    “A main weakness of the work is that the effect of the drug on virus yield (e.g. Fig 2, Fig. 7) is only about 3-fold or in some cases not much more than 1 log.”

The authors provide a new Fig 2 (which they call Fig. R1 for some reason), which shows statistical significance. Despite the author’s view that this new figure is “already dense” this must be shown to convince a reader that there are significant differences. The figure is not already dense, and this makes it a much more convincing figure. Thank you for including.

As to the differences in infectious virus versus glycoprotein expression, the authors respond back that the mechanism of action of the drug is a post-entry step, likely involving inhibition of RNA synthesis. While the post-entry step is clearly shown in the data in Figure 4 (and is well done), there is an assumption that this is at the level viral RNA which was not measured.

Thus, the text should add the caveat that viral RNA was not measured, and thus, the effect could be on translation or production of viral mRNA.

2)     The original Fig 6 did not include statistical analysis.

The authors now provide a new Fig. R2 with this analysis but claim not to want to use this version. Please include the new version with the statistics to improve the analysis and rigor of the study.

Other comments from this reviewer were adequately addressed.

Author Response

Response to Reviewer 2

“This revised manuscript by Sibille et al, as addressed most of the comments from this reviewer. There are still some points that must be addressed and modified Figures to include, as they are not completely clear to a reader”.

Response: We thank Reviewer 2 for her/his comments. We agreed with her/his suggestions to introduce some further improvements to make the paper more complete. Here below, there is a point-by-point response to the raised questions:

1) “A main weakness of the work is that the effect of the drug on virus yield (e.g. Fig 2, Fig. 7) is only about 3-fold or in some cases not much more than 1 log.” 

The authors provide a new Fig 2 (which they call Fig. R1 for some reason), which shows statistical significance. Despite the author’s view that this new figure is “already dense” this must be shown to convince a reader that there are significant differences. The figure is not already dense, and this makes it a much more convincing figure. Thank you for including. 

Response 1: As suggested by Reviewer 2, the Fig. R1 (Rebuttal Figure 1) previously included in the Response to Reviewer 2 for the ms. viruses-1906616.R1, has now been included as the new Figure 2 in the revised manuscript viruses-1906616.R2 (please, see page 7).

“As to the differences in infectious virus versus glycoprotein expression, the authors respond back that the mechanism of action of the drug is a post-entry step, likely involving inhibition of RNA synthesis. While the post-entry step is clearly shown in the data in Figure 4 (and is well done), there is an assumption that this is at the level viral RNA which was not measured.

Thus, the text should add the caveat that viral RNA was not measured, and thus, the effect could be on translation or production of viral mRNA”.

Response: According to the Reviewer 2’s advice, a new sentence stating that the levels of viral mRNAs were not measured, and so the MEDS433’s effect on HA and NA glycoprotein expression could be on mRNAs synthesis or on their translation has now been added at the end of the 3.3  paragraph in the Results section of the revised manuscript (please, see page 8, lines 503-505).

2) “The original Fig 6 did not include statistical analysis. 

The authors now provide a new Fig. R2 with this analysis but claim not to want to use this version. Please include the new version with the statistics to improve the analysis and rigor of the study”. 

Response 2: According to the Reviewer 2’s suggestion, the Fig. R2 (as for Rebuttal Figure 2) previously included in the Response to Reviewer 2 (ms. viruses-1906616.R1) has now been included as the new Figure 6 in the revised manuscript viruses-1906616.R2 (please, see, page 11).

“Other comments from this reviewer were adequately addressed”. 

Response: We thank Reviewer 2 for her/his accurate review of the manuscript and her/his criticisms and suggestions that allowed us to improve the quality of our study.

Reviewer 3 Report

I respect the authors for responding to all comments, including additional experiments, in such a short period of time. The manuscript is very improved. There is one point in the introduction part, and I think this paper should be accepted if that point is improved.

The authors described "Noteworthy, low frequencies of IV viruses with reduced susceptibility to NAIs have been reported in the past few years [10, 11]." (lines 42-44). However, this sentence does not connect to the later sentence "The available data on the ..." (lines 49-54) in terms of content. Therefore, it is necessary to describe the previous spread of oseltamivir-resistant virus following the sentence in lines 42-44.

Author Response

Response to Reviewer 3

“I respect the authors for responding to all comments, including additional experiments, in such a short period of time. The manuscript is very improved. There is one point in the introduction part, and I think this paper should be accepted if that point is improved.

The authors described "Noteworthy, low frequencies of IV viruses with reduced susceptibility to NAIs have been reported in the past few years [10, 11]." (lines 42-44). However, this sentence does not connect to the later sentence "The available data on the ..." (lines 49-54) in terms of content. Therefore, it is necessary to describe the previous spread of oseltamivir-resistant virus following the sentence in lines 42-44”.

Response: We thank Reviewer 3 for her/his comment. To address her/his request, language has been added in the Introduction section to describe in more detail the emergence of resistance to NAI among seasonal IAV during the 2007-2009 period, and the subsequent decrease to the current low levels (0.6% in 2019-2020) (Govorkova et al., Antivir. Res., 2022, 200, 105281) when the IAV (H1N1) pdm09 that did not carry the NA H275Y mutation when it emerged, became the dominant seasonal strain (please, see page 1, lines 42-25 and page 2, lines 66-69 in the revised manuscript).

In this regard, the rapid emergence of oseltamivir-resistant strains in 2007-2009 highlighted that NAI-resistance can develop and spread. This fact suggests the occurrence of a low genetic barrier against resistance to some anti-IV DAA and underlines the need for the development of new anti-IV agents with low-propensity to drive the emergence of viral resistance (Davidson, S. Front. Immunol. 2018, 9:1946).

I hope that through the more detailed description of the emergence of NAI resistance that it has been now included in the Introduction section of the revised manuscript, the reader can get a better logical connection with the subsequent sentence “The available data on the ….” that it has been slightly modified too (please, see page 2, lines 75-79 in the revised manuscript).

Reviewer 4 Report

The new revised version is much straightforward and easy to read and comprehend. All of my question has been resolved.

Author Response

Response to Reviewer 4 

“The new revised version is much straightforward and easy to read and comprehend. All of my question has been resolved”.

Response: We thank Reviewer 4 for her/his review, comments and suggestions that helped to improve the quality of our manuscript.